# Low Back Pain in Elderly from Belém-Pa, Brazil: Prevalence and Association with Functional Disability

**DOI:** 10.3390/healthcare9121658

**Published:** 2021-11-30

**Authors:** Elaine Miyuka Sato, Mauricio Oliveira Magalhães, Beatriz Coelho Jenkins, Lays da Silva Ferreira, Hallyson Andrey Raposo da Silva, Paulo Renan Farias Furtado, Eder Gabriel Soares Ferreira, Emmanuele Celina Souza dos Santos, Bianca Callegari, Amélia Pasqual Marques

**Affiliations:** 1Faculty of Physical and Occupational Therapy, Institute of Health Sciences, Federal University of Pará, Belém-Pará 66075-110, Brazil; biajenkinsc@gmail.com (B.C.J.); hassilva14@gmail.com (H.A.R.d.S.); paulorffurtado@gmail.com (P.R.F.F.); gabrielferreira_3@hotmail.com (E.G.S.F.); manusouza1469@gmail.com (E.C.S.d.S.); 2Laboratory of Human Motricity Sciences, Federal University of Pará, Belém-Pará 66050-160, Brazil; callegaribi@uol.com.br; 3Master’s Program in Human Movement Sciences, Federal University of Pará, Belém-Pará 66050-160, Brazil; mauriciomag20@gmail.com (M.O.M.); layssferreira@hotmail.com (L.d.S.F.); 4Department of Physiotherapy, Speech Therapy and Occupational Therapy, Faculty of Medicine, University of São Paulo, São Paulo 05508-060, Brazil; pasqual@usp.br

**Keywords:** aged, low back pain, prevalence, risk factors

## Abstract

Background: This cross-sectional study aimed to determine the prevalence of low back pain (LBP) in the elderly population living in Belém-Pará and to assess the spectrum of problems related to these diseases including the demographic, socioeconomic, occupational characteristics and disability in this population. Methods: Three structured questionnaires were applied in a randomly selected representative sample of 512 elderly people aged ≥60 years. Results: LBP prevalence in the elderly population was 55.7%. Among then, 56.1% had pain at the time of the interview (punctual prevalence), 91.7% had LBP in the last 365 days (prevalence in the last year), and 85.3% at some point in life (prevalence at some point in life). Overall, most studies are above average. LBP was positively associated with hypertension and the influence of the physical and mental health on their social activities ranged from slightly to extreme. LBP was negatively associated with characteristics, such as education (over 11 years), class A or B income, physical activity, high satisfaction with previous work, and excellent self-perceived health, corroborating to the literature. Conclusions: Greater intensity of pain and functional disability were associated with the presence of comorbidities, smoking habits, and low physical activity. LBP prevalence was high, above the national average, mainly affecting the underprivileged classes related to several modifiable factors, highlighting the importance of preventive and interventionist actions for healthy aging.

## 1. Introduction

Low back pain (LBP) is defined as pain or discomfort in the region below the costal margin and above the lower gluteal folds, with or without referred leg pain, which may be acute, subacute, or chronic. Nonspecific LBP is the most common type, with no relation to a known cause or pathology [1].

LBP is the main cause of disability worldwide and is associated with high costs for the health systems and for the individuals affected by direct costs, such as medical, diagnostic, and medication services, in addition to indirect costs reflected in reduced productivity and greater absenteeism at work [2].

Brazil has the second most prevalent chronic condition behind systemic arterial hypertension, and one of the main causes of interference in the quality of life, resulting in a significant increase in years lived with disabilities. The occurrence of such a condition accompanies population aging, directly correlated to it, since the physiological decline in the aging process makes the elderly more prone to chronic health conditions. Some LBP-related factors cannot be modified (e.g., gender, genetics), but others are modifiable (lifestyle and comorbidities) [3,4,5].

A recent systematic review has shown that there is a notable lack of studies with good methodological designs, using validated questionnaires capable of measuring the LBP prevalence in the Brazilian population. Most of them present a significant limitation, mainly in their methodological design related to external and internal validity, with a moderate to high risk of bias [6].

In the case of the elderly population, another systematic review with meta-analysis covered all studies conducted up to 2015 in Brazil, totaling only 16 studies covering cities in the Northeast, Midwest, South, and Southeast regions, highlighting the lack of studies in the northern region [7]. Recently, the first study including the North region was conducted in the older adult population in an Amazonia Brazilian community [8].

Despite being the second largest metropolitan city in the North Region of Brazil, with 2,275,032 inhabitants [3], there are no studies to date reporting the LBP prevalence in the elderly population in the city of Belém, PA.

Given the importance of the region’s representativeness (whereas Brazil is a country of continental dimension with great variability of population profile), the present study aimed to estimate the LBP prevalence in the elderly population living in the city of Belém-PA, in addition to substantiating demographic, socioeconomic, and occupational characteristics and disability related to LBP.

## 2. Methods

This epidemiological cross-sectional study was approved by the Ethics Committee of the Faculty of Medicine of the University of São Paulo, Brazil (#1581420). A total of 512 elderly volunteers, who signed the consent form, participated in the survey, in a convenience sample, through a structured interview in the municipality and metropolitan region of Belém, PA.

### 2.1. Participants and Sample Size

The population of this study was composed of elders of the municipality and metropolitan region of Belém, Pará, Brazil, aged 60 years or older, of both genders, without cognitive impairments to limit understanding and answering the questionnaire in Portuguese. The sample was enrolled by convenience and participants were excluded if they were unable to understand the questions presented; refused to sign the free and informed consent form; or had a self-reported clinical diagnosis of a serious condition in the spine, such as cancer, vertebral infection, vertebral compression fractures, cauda equina syndrome, ankylosing spondylitis, or neurodegenerative diseases [9].

For the sample size, the parameters used were the total elderly population in the city of Belém (*N* = 128,720) and average LBP global prevalence adjusted in the last month of 23.2% (*p* = 0.232) [10], 4% accuracy (*p* = 0.04), 95% confidence interval (z = 1.96), and maximum sample loss of 20%. Thus, the estimated population was 512 individuals.

### 2.2. Procedures

The assessment was administered through a structured interview based on three questionnaires, from May 2017 to May 2018, with questions being applied by a group of previously trained researchers. All scales and questionnaires used, which are described below, were translated and adapted to Brazilian Portuguese and their clinimetric properties tested [11,12,13].

To apply the questionnaires, the Google Forms survey administration application was used as part of the Google Drive office suite. Initially, the demographic, socioeconomic, anthropometric, behavioral, and occupational data of the participants were surveyed. The demographic, anthropometric, and socioeconomic variables recorded were gender, age, marital status, weight, height, body mass index (BMI), education level, and economic class.

Regarding the behavioral variables, the level of physical activity was assessed using the International Physical Activity Questionnaire short version [14], which classifies the level of physical activity by considering the frequency, duration, and intensity, stratifying the individual into sedentary (no physical activity for 10 continuous minutes), insufficiently active (light activities lasting 10 min for 5 days a week), active (moderate activities lasting more than 20 min for 3 to 5 days a week), and very active (vigorous activities lasting more than 30 min and for more than 5 days a week) [15]. In addition, smoking was also considered, dividing respondents into non-smokers (those who never smoked), ex-smokers (those who quit more than a year ago), and current smokers (those who consumed any number of cigarettes per day) [16].

Regarding occupational characteristics, we asked about satisfaction in previous jobs, considering those satisfied who felt well fulfilled in their profession and those dissatisfied who did not feel fulfilled at work, and whether the elderly still performed any work at the time of the interview. In the second questionnaire, the volunteers answered if they had LBP in any of three different periods: at the time of the interview (punctual), in the last 365 days (last year), and at any time in life (some time in life). For the purpose of conceptual standardization, participants received an illustrative figure of the human body, specifying the lumbar region (Figure 1) and were instructed to consider an episode of pain “any pain between the last rib and the bottom of the buttocks lasting more than 24 h, preceded by 30 days without pain” [17]. In the affirmative answers, information was also collected on frequency, intensity, duration, and irradiation of pain to the leg [10], in addition to the average pain intensity, quantified using the numeric rating scale (NRS), a scale of 11 points, ranging from 0 to 10, with zero indicating no pain and 10 the worst pain imaginable [13,17].

Finally, in a third questionnaire, the participants who reported pain were given the Roland–Morris disability questionnaire [12,18,19], composed of 24 items that describe everyday situations in which the subjects may have difficulty performing because of LBP. Each affirmative answer corresponds to a point, and the final score ranges from 0 (the absence of disability) to 24 (severe disability).

### 2.3. Statistical Analysis

The quantitative variables were expressed by the mean and standard deviation, and the qualitative variables were expressed as absolute frequency and percentage. To verify the association between qualitative variables, univariate logistic regression was used to express the effect of the odds ratio (OR). For the comparison of quantitative variables between the two groups, the Student *t*-test or Mann–Whitney test was used depending on the assumption of normality of the data, and in cases of three or more groups, ANOVA (assumption of normality) or the Kruskal–Wallis test (non-parametric). If there was significance in any of these tests, a comparison was made in pairs to determine which groups differed by Turkey test. Spearman’s correlation between quantitative and ordinal variables was calculated. The level of significance was set at α < 0.05. Statistical tests were performed using R core (version 3.6.1, R Foundation, Vienna, Austria) and SPSS (version 19, IBM, Armonk, NY, USA). There were no missing data.

## 3. Results

A total of 512 elderly individuals participated in the study with a mean age of 70.4 ± 7.4 years and mean BMI 26.1 ± 4.2. Among these, the majority were women (69.7%), mixed race (41.2%), married (37.9%), sedentary (38.7%), and had no income or class E income (62.3%). Furthermore, only 53 (10.4%) reported dissatisfaction with their work throughout their lives.

LBP prevalence was 55.7% (285), with a mean pain of 5.7 ± 2.3; 160 respondents (56.1% of the 285) accused pain at the time of the interview (punctual prevalence), 261 (91.7%) in the previous 365 days (prevalence in the last year), and 243 (85.3%) at some point in life (prevalence at some point in life).

Regarding the clinical characteristics of the participants with pain at the time of the interview, 72 (45%) elderly said that this pain was enough to limit their usual activities and 99 (61.9%) felt the pain radiating to the leg. Regarding the participants who reported pain in the last year, 157 (60.2%) stated that this pain was sufficient to limit their usual activities and 168 (64.4%) felt the pain radiating to the leg. In addition, among participants who reported pain at some point in life, 162 (66.7%) stated that this pain was sufficient to limit their usual activities, and 159 (65.4%) felt the pain radiating to the leg.

Regarding individual characteristics, there was a positive and significant association between LBP and hypertension (HT) (OR, 2.1; 95% CI, 1.5–3.1) and the influence of physical and mental health interfered slightly (OR, 3.6; 95% CI, 2.3–5.7) to extremely (OR, 11.1; 95% CI, 1.3–91.3) in their social activities. Education (>11 years; OR, 0.3; 95% CI 0.2–0.6), income (classes A or B; OR, 0.3; 95% CI 0,1–0.7), physical activity (very active; OR, 0.3; 95% CI 0.1–0.5), satisfaction with their previous work (very satisfied; OR, 0.2; 95% CI 0.1–0.3), and self-perception of health (excellent; OR, 0.1; 95% CI 0.0–0.4) showed significant negative associations with reported LBP (Table 1).

Table 2 summarizes the correlations between population characteristics and the levels of pain and functional disability. For variables with more than two categories, post hoc analysis was performed. 

We identified that pain intensity (score ranges from 0, the absence of pain, to 10, worse pain) was higher in women (*p* < 0.02), in individuals with osteoarthritis (*p* < 0.05), and in blacks (*p* < 0.01). 

A greater functional disability (score ranges from 0, the absence of disability, to 24, severe disability) was identified in the elderly who presented HT (*p* < 0.01), had diabetes mellitus (*p* < 0.01), underwent less than 11 years of education (*p* < 0.01), had brown skin (*p* < 0.01), were widowed (*p* < 0.01), were sedentary (*p* < 0.01), and were ex-smokers (*p* < 0.01). Moreover, elderly people who consumed alcohol once a month or less (*p* < 0.03), were classified in class D income or below (*p* < 0.01), were more dissatisfied with previous work (*p* < 0.01), had poor or regular self-perceived health (*p* < 0.01), and any negative influence of physical and mental health on social activity (*p* < 0.01) (Table 2). There were no missing data.

## 4. Discussion

This is the first representative study with the objective of measuring the LBP prevalence in the elderly population in the city of Belém, PA. Prevalence rates showed LBP to affect more than half of the elderly, leading to significant functional disability. The LBP prevalence was measured in three moments, being 56.1% at the time of the interview, 91.7% in the last year, and 85.3% at any moment in life. Between 45–66.7% of the elderly stated that LBP limited their usual activities, and 61.9–65.4% felt the pain radiating through the leg. HT and the negative influence of physical and mental health on social activities are positively associated with LBP, in contrast to extended education, higher income, levels of physical activity, satisfaction with previous work, and better self-perception of health, which were identified as negatively associated with LBP.

The prevalence of punctual LBP in Belém (56.1%) was above the punctual prevalence in Brazil (25%) [7]; however, there is no homogeneity in the literature regarding the LBP definitions and evaluated regions. For this reason, the worldwide prevalence has a wide spectrum ranging from 21% to 75% among the elderly populations [20]. The prevalence in the last year, found to be 91.7%, was much higher than the 13% systematized in the review by Leopoldino et al. [7]. However, a more recent systematic review identified annual prevalence rates of 21.7–68.3% [20], including three Brazilian studies ranging between 55.8–68.3% [21,22,23]. In addition, the prevalence at some point in life (85.3%) is high and similar to the values estimated in industrialized countries [1]. In comparison to another study in the north region (Manaus, Amazonas), we found similar prevalence, with punctual prevalence was 42.4% and the prevalence for the last 365 days was 93.7% [8].

In the present study, having HT and declaring that physical and mental health negatively influences social activities was positively associated with LBP. Quintino et al. [23] also found a greater propensity to pain in individuals with more severe comorbidities, especially in those with more than three. In a recent study, De Souza et al. [8] found association between LBP and BMI, health perception, and emotional level in a similar population in Manaus, Amazonas.

Other characteristics, such as the extent of education, income, physical activity, satisfaction with previous work, and good self-perception of health, was negatively associated with pain. These results corroborate with those reported by Barros et al. [24], who evaluated the sociodemographic characteristics of 1518 elderly people from Campinas, SP and found that elderly people with more than nine years of education were more active, with better self-reported health and lower comorbidities, for example, hypertension and LBP. In addition, a review by Wong et al. [5] found that less educated elderly people with lower income, smokers, and females, in addition to those with psychological symptoms, limiting beliefs about pain, who performed vigorous physical activities and had poor self-reported health and other comorbidities, were more prone to LBP.

In our study, being female was not directly related to the presence of pain, and this difference can also be observed in the literature [22,25]; however, this characteristic is not described as a direct cause of pain, but as a population that may be more prone to hormonal, psychological, or social causes [26]. Smoking was evidenced as a risk factor for the development and worsening of chronic pain; however, this association was not significant in this study and may have been masked by the low prevalence of smokers among the elderly people who we interviewed [27]. Elderly people who presented with osteoarthritis, who were female, and who were black had greater pain intensity. In the study by Pereira et al. [28], which included 934 elderly people, it was observed that a higher LBP intensity was associated with worse self-perception of health, the presence of comorbidities, joint diseases, and female gender, similar to our findings. The female gender factor in the increased perception of pain intensity may be related to greater sensitivity because of biopsychosocial contributions, such as the influence of hormonal reduction after menopause and its greater tendency to catastrophize pain [26,29].

The association between greater pain intensity and osteoarthritis may be because of its greater co-occurrence as age advances or to the inflammatory contributions of joint wear; however, it is not usually seen as an expressive portion of asymptomatic patients, highlighting the complexity of the pain mechanism [30].

The association between pain and race is not yet well established. Indeed, the pain seems to be more associated with socioeconomic factors, such as education, income, and exposure to risks, with the black population of our country being more expressive in the less favored classes [31,32].

In addition, elderly people that presented higher functional disability had HT, had DM, underwent less than 11 years of schooling, were mixed race, were widowed, were sedentary, were an ex-smoker, consumed alcohol once a month or less, were in income class D or lower, were dissatisfied with the work performed, had poor or regular self-perception of health, and were subject to negative influence of physical and mental health on a social activity.

Low back pain, when accompanied by irradiation to regions below the knee, was highly observed in this study. This demonstrates a warning factor, since it can be related to pathologies that lead to radicular involvement, and can represent a group with higher functional disability, pain intensity, and, consequently, worse prognosis, when compared with patients who present only a local low back pain [33]. Studies indicate that pain, such as low back pain, affects elderly in the basic and instrumental activities of daily living [25]. The characteristics of the elderly with higher functional disability found in this study corroborate with the literature. Some studies show that the Brazilian elderly population with low education, low income, worse self-rated health, and chronic diseases have high scores of functional disability associated [8,34]. More specifically, elderly people with low back pain, or sedentary elderly people with more comorbidities, have higher functional disability [22].

Thus, the importance of sociodemographic characteristics and health indicators on pain and functional disability is evidenced, as higher income and education are directly related to greater and better access to health services, access to information and the viability of a healthier lifestyle, and better working conditions, representing less exposure to occupational risks. Moreover, in view of the growth projections in the number of elderly people worldwide, knowing the prevalence of low back pain is extremely important to improve the management of this condition, which is still precarious. This involves optimizing and minimizing expenses with highly complex problems (for the person and for the country), investing in public policies of prevention, and providing information and earlier interventions.

The results of this study, due to its cross-sectional design, should be interpreted with caution, visualized as representing a situation characteristic of a given moment, with a construction of the causality relationship limited by the knowledge of the event schedule. The inquiries due to their automatic reported character are influenced by memory, but they still consider that pain and disability are subjectively assessed for suffering socio-cultural influences due to their biopsychosocial character.

## 5. Conclusions

In this study of the elderly population of Belém, PA, the LBP prevalence was high (56.1%), above the national average (25%), mainly affecting the underprivileged classes. Greater functional disability correlated with several modifiable factors, highlighting the importance of education in changing the situation by gaining access to better jobs, and, thus, better income; less exposure to occupational risks; better access to information; and an ability to adopt healthy habits, physical exercise, and healthcare. Thus, this study highlights the importance to increase qualified studies on the subject in a large scale in the country by improving investments in public policies to manage pain in the elderly population, in addition to preventive actions throughout the entire life to minimize the impact of the lack of access to information and services on healthy aging.

## Figures and Tables

**Figure 1 healthcare-09-01658-f001:**
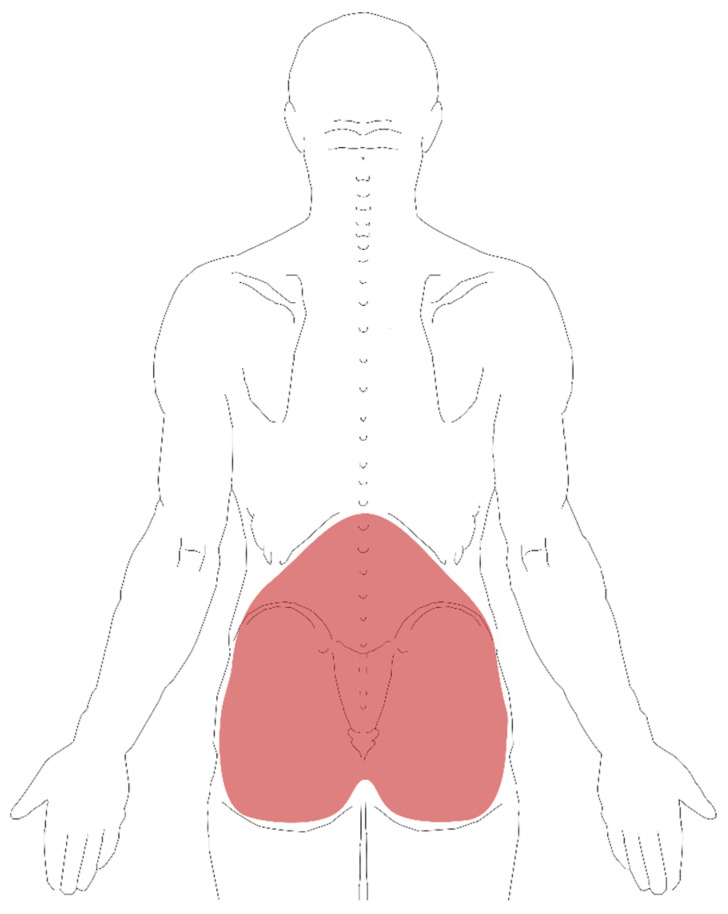
Low back pain map.

**Table 1 healthcare-09-01658-t001:** Characteristics of the elderly population in the metropolitan region of Belém, PA.

Outcome		Do You Have or Did You Have Low Back Pain				O.R.	IC(95%) Do O.R.		*p*-Value
		No		Yes					
		*N*	%	*N*	%		Inf. Lim.	Sup. Lim.	
Gender	Female	152	67.0%	205	71.9%	1.000			
	Male	75	33.0%	80	28.1%	0.791	0.542	1.155	0.2244
Race	Yellow	12	5.3%	9	3.2%	1.000			
	White	105	46.3%	101	35.4%	1.283	0.518	3.175	0.5905
	Mixed	80	35.2%	131	46.0%	2.183	0.881	5.413	0.0919
	Black	30	13.2%	44	15.4%	1.956	0.733	5.216	0.1802
Marital Status	Single	49	21.6%	53	18.6%	1.000			
	Stable Union or Married	100	44.1%	119	41.8%	1.100	0.687	1.762	0.6910
	Divorced	18	7.9%	36	12.6%	1.849	0.931	3.673	0.0790
	Widow	60	26.4%	77	27.0%	1.186	0.709	1.985	0.5150
SAH	No	152	67.0%	139	48.8%	1.000			
	Yes	75	33.0%	146	51.2%	2.129	1.483	3.055	<0.0001
DM	No	182	80.2%	209	73.3%	1.000			
	Yes	45	19.8%	76	26.7%	1.471	0.967	2.236	0.0710
Arthrosis	No	194	85.5%	227	79.6%	1.000			
	Yes	33	14.5%	58	20.4%	1.502	0.940	2.399	0.0887
Arthritis	No	217	95.6%	269	94.4%	1.000			
	Yes	10	4.4%	16	5.6%	1.291	0.574	2.902	0.5370
Degree of Education	0–4 years	48	21.1%	97	34.0%	1.000			
	5–8 years	39	17.2%	41	14.4%	0.520	0.298	0.909	0.0220
	9–11 years	83	36.6%	109	38.2%	0.650	0.415	1.018	0.0600
	>11 years	57	25.1%	38	13.3%	0.330	0.193	0.564	<0.0001
Income	None ou Class E	130	57.3%	189	66.3%	1.000			
	Class D	44	19.4%	61	21.4%	0.954	0.610	1.492	0.8350
	Class C	34	15.0%	26	9.1%	0.526	0.301	0.918	0.0240
	Class B or A	19	8.4%	9	3.2%	0.326	0.143	0.743	0.0080
Physical Activity	Sedentary	58	25.6%	140	49.1%	1.000			
	Insufficiently active	41	18.1%	59	20.7%	0.596	0.361	0.985	0.0436
	Active	102	44.9%	70	24.6%	0.284	0.185	0.438	<0.0001
	Very active	26	11.5%	16	5.6%	0.255	0.127	0.510	0.0001
Smoking	No	128	56.4%	150	52.6%	1.000			
	Ex	93	41.0%	123	43.2%	1.129	0.789	1.614	0.5077
	Yes	6	2.6%	12	4.2%	1.707	0.623	4.676	0.2986
Alcohol Consumption How often do you consume alcohol-containing drinks	Do not applicable	162	71.4%	207	72.6%	1.000			
	Once a month or less	36	15.9%	42	14.7%	0.913	0.559	1.491	0.7160
	Twice to four times a month	22	9.7%	28	9.8%	0.996	0.549	1.806	0.9900
	Twice or more times a week	7	3.1%	8	2.8%	0.894	0.318	2.518	0.8330
Satisfaction	Dissatisfied	11	5.0%	42	14.9%	1.000			
	Satisfied	136	61.5%	193	68.7%	0.372	0.185	0.748	0.0055
	Very Satisfied	74	33.5%	46	16.4%	0.163	0.076	0.348	<0.0001
Are you currently working	No	186	81.9%	224	78.6%	1.000			
	Yes	41	18.1%	61	21.4%	1.235	0.795	1.920	0.3474
During the past four weeks. how did your physical health or emotional problems interfere with your normal social activities in relation to family, friends or group	No way	150	66.1%	95	33.3%	1.000			
	Lightly	38	16.7%	86	30.2%	3.573	2.255	5.662	<0.0001
	Moderately	29	12.8%	72	25.3%	3.920	2.373	6.475	<0.0001
	Quite	9	4.0%	25	8.8%	4.386	1.963	9.801	0.0003
	Extremely	1	0.4%	7	2.5%	11.053	1.339	91.255	0.0257
Do you classify your health as	Poor	2	0.9%	15	5.3%	1.000			
	Fair	64	28.2%	131	46.0%	0.273	0.061	1.230	0.0909
	Good	114	50.2%	107	37.5%	0.125	0.028	0.560	0.0066
	Very Good	22	9.7%	18	6.3%	0.109	0.022	0.541	0.0067
	Excelent	25	11.0%	14	4.9%	0.075	0.015	0.375	0.0016
Age (years)		71.38	7.69	69.61	7.07	0.968	0.945	0.991	0.0076
BMI (kg/m^2^)		25.62	4.05	26.45	4.24	1.050	1.006	1.096	0.0259
BMI	Malnutrition	38	16.7%	35	12.3%	1.000			
	Eutrophy	114	50.2%	138	48.4%	1.314	0.780	2.215	0.3050
	Obesity	75	33.0%	112	39.3%	1.621	0.941	2.794	0.0820

O.R.: oddis ratio; IC (95%): interval of 95% de confidence; SAH: systemic arterial hypertension; DM: diabetes mellitus; BMI: body mass index.

**Table 2 healthcare-09-01658-t002:** Population characteristics and the level of pain and functional disability.

Outcome		Pain Scale			Sum (RM)		
		Mean	Standard Deviation	*p*-Value	Mean	Standard Deviation	*p*-Value
Gender	Female	5.89	2.33	0.0272	9.78	7.62	0.1626
	Male	5.18	1.95		11.09	8.34	
SAH	No	5.84	2.38	0.3212	7.70	6.91	<0.0001
	Yes	5.55	2.12		12.49	7.95	
DM	No	5.62	2.39	0.4775	8.86	7.32	<0.0001
	Yes	5.88	1.83		13.68	8.15	
Arthrosis	No	5.55	2.20	0.0474	10.40	7.52	0.2465
	Yes	6.22	2.38		9.16	8.96	
Arthritis	No	5.69	2.24	0.9634	10.20	7.89	0.6438
	Yes	5.63	2.55		9.19	7.04	
Degree of Education	0–4 years	5.82	2.46	0.4233	9.61	7.55	0.0004
	5–8 years	6.12	2.51		10.51	7.48	
	9–11 years	5.47	2.05		12.06	8.08	
	>11 years	5.50	1.94		5.61	6.26	
Race	Yellow	5.22	2.95	0.0028	7.00	8.23	0.0002
	White	5.74	2.60		7.51	7.36	
	Mixed	5.34	1.89		11.96	7.65	
	Black	6.68	1.99		11.41	7.73	
Marital Status	Single	5.79	2.31	0.7251	7.74	7.00	0.0009
	Stable Union or Married	5.82	2.16		10.00	8.08	
	Divorced	5.64	1.93		14.86	6.85	
	Widow	5.44	2.50		9.82	7.62	
Physical Activity	Sedentary	5.62	1.99	0.1642	12.45	7.84	<0.0001
	Insufficiently active	5.86	2.00		9.97	7.74	
	Active	5.94	2.81		6.76	6.38	
	Very active	4.50	2.34		5.44	6.86	
Smoking	No	5.71	2.43	0.6607	8.45	7.20	0.0002
	Ex	5.63	2.04		12.26	7.97	
	Yes	6.08	2.07		9.58	9.37	
Alcohol Consumption How often do you consume alcohol-containing drinks	Do not applicable	5.64	2.34	0.2722	10.32	7.81	0.0077
	Once a month or less	5.67	1.95		12.19	7.13	
	Twice to four times a month	6.21	1.97		7.43	8.27	
	Twice or more times a week	5.13	2.59		4.25	5.99	
Are you currently working	No	5.78	2.30	0.1032	10.65	7.97	0.0861
	Yes	5.34	2.04		8.30	7.05	
Income	None ou Class E	5.85	2.31	0.3809	10.58	7.88	0.0030
	Class D	5.20	1.98		11.49	7.70	
	Class C	5.73	2.65		5.23	5.40	
	Class B or A	5.56	1.13		6.00	8.19	
Satisfaction	Dissatisfied	5.50	2.14	0.8732	14.10	7.91	0.0002
	Satisfied	5.70	2.25		10.09	7.69	
	Very Satisfied	5.72	2.48		6.61	6.96	
During the past four weeks, how did your physical health or emotional problems interfere with your normal social activities in relation to family, friends or group	No way	5.82	2.58	0.3214	5.00	5.70	<0.0001
	Lightly	5.31	2.19		11.66	7.19	
	Moderately	5.69	1.90		13.58	7.66	
	Quite	6.24	2.09		14.32	7.38	
	Extremely	6.43	1.99		11.00	9.09	
Do you classify your health as	Poor	7.00	2.00	0.1264	17.47	6.33	<0.0001
	Fair	5.83	1.89		14.08	6.85	
	Good	5.35	2.62		6.49	6.32	
	Very Good	5.83	2.23		3.83	4.90	
	Excelent	5.36	2.21		1.50	2.71	
BMI	Malnutrition	5.30	2.75	0.4190	6.50	6.40	0.0958
	Eutrophy	5.78	2.24		7.58	6.83	
	Obesity	5.67	2.24		11.48	8.00	

RM: Roland–Morris; SAH: systemic arterial hypertension; DM: diabetes mellitus; BMI: body mass index.

## Data Availability

The data presented in this study are available on request from the corresponding author. The data are not publicly available due to personal participation information.

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
