# Peer review of "Low Back Pain in Elderly from Belém-Pa, Brazil: Prevalence and Association with Functional Disability"

_healthcare, 2021, doi:10.3390/healthcare9121658_

Round 1

Reviewer 1 Report

Introduction:

What does this region have in particular to explain the importance of doing a study there (number of inhabitants? First nations? Special geographic situation? Etc.), I conclude that this region is not part of the regions mentioned in line 55?

Is the problem of a methodological nature (invalid questionnaires?) Mentioned in lines 50-52?

Studies up to 2015 start from what year and what did these studies cover and what did not cover? Did the authors want to clarify the few studies in elderly people with low back pain? What is the logical connection with the aim of this study?

The introduction should provide sufficient background and be better articulated to emphasize the relevance of the aim of this study.

Method

Sample Size  randomly selected, i.e.?

Line 72: In general, elderly is from 65 years old, why were the 60 year old people included in this sample?

Table 2 is difficult to read / understand. A sction wiht more details of descriptive results on functional disability should help to better track the correlation between disability and other parameters in Table 2.

Discussion

Line 180-181: «This is the first representative study with the objective of measuring the LBP prevalence in the elderly population of the city of Belém-PA». The calculation of the sample size mentioned in lines 79-80 seems to indicate that an epidemiological surveillance of low back pain is in progress in this city; this study would therefore not be the first although transversal and although in elderly.

«In this representative study of the elderly population of Belém-PA, the LBP prevalence was high, above the national average».  What is the national average for lower back pain in the elderly?

Conclusion

This paper has the merit of providing a respectable number of parameters and up-to-date data; however, there is nothing new. The fact remains that this paper confirms what is already established among the elders. The authors should tell the reader what the contribution of this study is to the scientific literature already available.

Reviewer 2 Report

Dear Authors

Thank you for the opportunity to review your article.

Brief summary: This is a This epidemiological cross-sectional study that aims to determine the prevalence of low back pain (LBP) in the elderly population living in Belém-Pará and to assess the associated factors of the LBP in this population.

Areas of strength

There is strong concordance between the and the methods used. The description of the methodology was made in a clear and adequate way, some information is missing. The results are clearly described. The discussion correlates with the presented data and takes the published literature into account. The manuscript presents some limitations.

Weakness:

  1. The bibliographical references included are relevant to the topic under study but only present 9/34 (26%) references in the last 5 years.
  2. The references follow the style of referencing adopted by the journal. Please consult the guidelines for authors.
  3. Page 1, line 18 – (…) related to these diseases related to the (…) - please review
  4. Page 2 lines 66-68 – “ A total of 512 randomly selected elderly volunteers, who signed the consent form, participated in the survey, in a convenience sample, (…)” It is not clear how a randomly selected sample is identical to a convenience sample? In a convenience sample there is no randomization. Therefore it is not representative as stated in the conclusion. Please review.
  5. Page 2, lines 83-124 in the section “2.2. Procedures” please, specify the clinometric properties of the tools used.
  6. Page 4, line 132-133 “If there was significance in any of these tests, a comparison was made in pairs to determine which groups differed. “ please indicate the test post hoc used
  7. Page 11, line 249 – “ daily and instrumental activities of life [25].” Please review. it will be “basic and instrumental activities of daily living”?
  8. Page 11 line 255-258 - Please reinforce the practical implications of your study findings.
  9. Page 12, line 268 – “corelated” it will be “correlated”?

Round 2

Reviewer 2 Report

Dear Authors

Thanks for the opportunity to review the article again. Congratulations, the article has been improved and its quality increased.

In the manuscript, the bibliographic references in the last 5 years increased from 26% to 33% (11/34).

Aspects to correct

Page 2 , line 60 “Amazonia Brazilian community [8],” correct please to “Amazonia Brazilian community [8].”

Line 314 to 399. The references follow the style of referencing adopted by the journal. Please consult the guidelines for authors

Example:

MDPI and ACS Style

Meltzer, G.Y.; Chang, V.W.; Lieff, S.A.; Grivel, M.M.; Yang, L.H.; Jarlais, D.C.D. Behavioral Correlates of COVID-19 Worry: Stigma, Knowledge, and News Source. Int. J. Environ. Res. Public Health 202118, 11436. https://doi.org/10.3390/ijerph182111436